# *Arabidopsis* WRKY53, a Node of Multi-Layer Regulation in the Network of Senescence

**DOI:** 10.3390/plants8120578

**Published:** 2019-12-06

**Authors:** Ulrike Zentgraf, Jasmin Doll

**Affiliations:** Center for Plant Molecular Biology (ZMBP), University of Tuebingen, Auf der Morgenstelle 32, 72076 Tuebingen, Germany; jasmin.doll@zmbp.uni-tuebingen.de

**Keywords:** leaf senescence, WRKY transcription factors, WRKY53, mechanisms of network regulation, redox regulation

## Abstract

Leaf senescence is an integral part of plant development aiming at the remobilization of nutrients and minerals out of the senescing tissue into developing parts of the plant. Sequential as well as monocarpic senescence maximize the usage of nitrogen, mineral, and carbon resources for plant growth and the sake of the next generation. However, stress-induced premature senescence functions as an exit strategy to guarantee offspring under long-lasting unfavorable conditions. In order to coordinate this complex developmental program with all kinds of environmental input signals, complex regulatory cues have to be in place. Major changes in the transcriptome imply important roles for transcription factors. Among all transcription factor families in plants, the NAC and WRKY factors appear to play central roles in senescence regulation. In this review, we summarize the current knowledge on the role of WRKY factors with a special focus on WRKY53. In contrast to a holistic multi-omics view we want to exemplify the complexity of the network structure by summarizing the multilayer regulation of WRKY53 of *Arabidopsis*.

## 1. Introduction

Degreening of leaves is the visible part of the senescence program. Chlorophyll is broken down and the photosynthetic apparatus is dismantled leading to light green and yellowish leaves. However, when these signs become apparent, the senescence program got into gear long before and molecular changes were already realized on several levels. The cells are able to integrate a plethora of signals, which drive onset and progression of senescence and even a reversal of the already started program is possible. The main driving force of developmental senescence is the age of the leaves and the age of the whole plant; however, we still do not know how the plant senses these parameters. Almost all plant hormones can influence the senescence program in synergistic or antagonistic ways. Small molecules like hydrogen peroxide or other reactive oxygen species (ROS) or intracellular Ca^2+^ levels can also act as signaling molecules; here, the question how specificity is achieved is still open. Remarkably, for intracellular H_2_O_2_, a long-term increase over several days at bolting and flowering time can be measured in *Arabidopsis* and oilseed rape [1,2], which appears to be different to oxidative bursts in stress responses within minutes and hours. Moreover, intracellular localization of ROS production appears to be important, as the hydrogen peroxide levels in different compartments contribute to a different extent to the senescence promotion [2] (unpublished results). However, the most obvious changes are the massive reprogramming of the transcriptome. In *Arabidopsis*, several thousand genes are up- or down-regulated during onset and progression of senescence. Detailed transcript profiling over 22 time points of a defined leaf of *Arabidopsis thaliana* (*At*) during onset and progression of leaf senescence enabled to build up a distinct chronology of events [3]. The first genes to be activated are related to autophagy and transport processes followed by genes related to production and scavenging of ROS. Subsequently, genes related to abscisic acid (ABA) and jasmonic acid (JA) production and signaling are induced indicating that ROS, ABA and JA are important early signals in leaf senescence. This is in agreement with a relatively early increase in JA [3] and the even earlier increase of intracellular hydrogen peroxide contents during bolting and flowering of *Arabidopsis* plants, which coincides exactly with the time point when monocarpic senescence is induced [1]. Remarkably, lowering hydrogen peroxide levels in *Arabidopsis* plants delayed the onset of leaf senescence [2].

These massive transcriptional changes imply a central role for transcription factors. Almost all transcription factor families in plants are involved in senescence regulatory processes; however, the families of WRKY and NAC factors, which largely expanded in the plant kingdom, are overrepresented in the senescence transcriptome of *Arabidopsis* [4]. Not only in *Arabidopsis* but also in other plant species, members of both families play important roles in senescence regulation [3,5,6,7,8,9,10,11,12,13]. This review will concentrate on the WRKY factors; an overview on the WRKY factors upregulated in expression (Table 1) or directly regulating senescence is presented (Table 2). A special focus is given to *At*WRKY53, as this factor appears to be the family member in *Arabidopsis* whose role in senescence has been analyzed in most detail. *At*WRKY53 acts as one of the regulatory hubs and, therefore, we would like to use this factor as an example to demonstrate the complexity of the WRKY network, which is part of the senescence regulatory network. If not indicated otherwise, in the following we portray regulatory mechanisms in *Arabidopsis*.

At the beginning of this century, the first connections between WRKY factors and senescence have been established in *Arabidopsis* [27,28,51,52]. After the finalization of the *Arabidopsis* genome project, it became clear that 75 WRKY transcription factor genes exist, which can be categorized into three different groups based on the presence and number of their protein motifs and domains [51,53]. All factors contain at least one DNA-binding domain consisting of the highly conserved and name giving WRKYGQK motif at the N-terminus and a zinc-finger structure at the C-terminus. This zinc-finger structure can be formed by either Cx_4-5_Cx_22-23_HxH (C2H2) or Cx_7_Cx_23_HxC (C2HC) and one zinc atom interacts with the cysteine and histidine residues to generate a finger-like structure. Group I factors contain two DNA-binding domains with two C2H2 zinc-finger structures, while Group II and III WRKY factors have only one DNA-binding domain with a C2H2 and a C2HC zinc-finger structure, respectively. The WRKYGQK motif as well as the zinc-finger structure are inevitable for the DNA binding of WRKYs as mutations in both structures reduced or even completely abolished the DNA binding [54]. However, small sequence variation in the highly conserved WRKYGQK motif and in the zinc-finger structures have been described in tomato or barley plants [55,56]. In general, the WRKY DNA-binding domain recognizes the consensus binding-motif TTGACC/T called W-box. Pathogen attack as well as abiotic stress conditions can activate transcription and/or activity of many WKRY factors (for review see [57,58]). However, there appears to be a large overlap of gene induction during senescence, many WRKYs are also activated during this developmental phase. Members of all three groups are part of the senescence regulatory network and many of these react to ROS, SA and JA signals.

Remarkably, the existence of one or more W-boxes in the promoters of almost all WRKY genes of *Arabidopsis* point to a WRKY transcriptional network, in which WRKYs regulate the expression of each other. This also implies internal feedback loops, as several but not all WRKYs regulate the expression of their own genes [59,60]. In this sense, not all WRKYs bind to all W-boxes; the surrounding sequences and/or other factors influence binding capability [61] so that a certain specificity of each W-box for a subset of WRKYs appears to exist. In addition, WRKY factors can also form heterodimers, which can have other DNA-binding preferences than homodimers [62,63]. This already makes it clear that we have a complex interplay of different WRKY factors in a WRKY network structure, e.g., WRKY18 acts as upstream regulator, downstream target and protein interaction partner of WRKY53, and vice versa [63]. Furthermore, a plethora of other proteins interacts physically with WRKY proteins influencing their activity and stability (for review see [64]).

## 2. Chromatin Remodeling: SUV2, POWERDRESS/HDA9, HAC1, JMJ16

WRKY53 is one of the central hubs in the WRKY network regulating early senescence. This WRKY factor is tightly regulated by multi-layer mechanisms to control its gene expression and protein stability and activity. First, the chromatin region has to be prepared for gene expression. While cytosine methylation directly on the DNA remains unchanged and overall very low during onset and progression of senescence in the *WRKY53* promoter [65], di- and tri-methylation of HISTONE3 (H3K4me2 and H3K4me3) in the promoter region of *WRKY53* increases and makes the DNA more accessible for transcription [66,67]. In contrast, the JmjC-domain containing protein JMJ16, which is a specific H3K4 demethylase, negatively regulates leaf senescence at least partly through repressing the expression of *WRKY53* [68]. This suggests that the accessibility of the chromatin at the *WRKY53* promoter region can be driven back and forth, most likely to adjust *WRKY53* expression and senescence induction to environmental conditions. Moreover, the single strand-binding protein WHIRLY1 binds to the opened promoter of *WRKY53* and represses the enrichment of H3K4me3, but enhances the enrichment of H3K9ac at this gene locus [69]. On top of that, the WRKY53 protein can also directly interact with chromatin modifiers like HISTONE DEACETYLASE9 (HDA9) mediating the recruitment of the POWERDRESS/HDA9 complex to W-box containing promoter regions of key negative senescence regulators. There, H3 acetylation marks are removed and thereby suppress the expression of these genes and foster senescence [70]. This indicates that WRKY53 is part of a positive feedback loop also on chromatin level. Very recently, WRKY53 and WRKY18 have been identified to regulate the expression of sugar response genes by recruitment of HISTONE ACETYLTRANSFERASE 1 (HAC1) to facilitate the acetylation of histone 3 Lys 27 (H3K27ac) on their promoters [71]. 

## 3. Regulation of *WRKY53* Expression

In addition to chromatin structure, binding of specific transcriptional activators or repressors influence the transcription rate of *WRKY53*. Many different proteins, at least 12, most likely even more, are able to bind to the promoter of *WRKY53* (GATA4, AD-Protein, WRKY53 itself, several other WRKYs, MEKK1, REVOLUTA, WHIRLY1) and influence its expression [8,63,65,72,73,74,75]. This includes not only canonical transcription factors, such as GATA4 or the HD-ZIPIII factor REVOLUTA, but also more atypical DNA-binding proteins. The MAP kinase kinase kinase MEKK1 has been identified as a DNA-binding protein of the *WRKY53* promoter and the binding region appears to be involved in the switching of the *WRKY53* expression from a leaf-age- to a plant-age-dependent expression even though the protein itself has no own activation domain [52,72]. The AD-protein contains, in fact, a transcriptional activation domain but also a kinase domain with similarities to HPT kinases. Autophosphorylation of AD protein can increase its DNA-binding activity towards the *WRKY53* promoter and can positively regulate *WRKY53* expression. However, the AD-protein is also able to interact physically with MEKK1 so that a competition for binding of MEKK1 between WRKY53 and the AD-protein might take place at the *WRKY53* promoter [73]. The single-stranded DNA-binding protein WHIRLY1 represses the expression of *WRKY53* and is shifted from the chloroplast to the nucleus by phosphorylation through CALCINEURIN B-LIKE-INTERACTING PROTEIN KINASE14 [75,76]. Moreover, accessibility of the promoter region most likely due to chromatin remodeling can change the outcome of the DNA-binding. If, for example, 1 kbp or 2.8 kbp upstream of the *WRKY53* coding region were used in GUS assays, the influence of effector proteins was different; in extreme cases, effectors had the opposite effect when interacting with the short promoter [63]. Therefore, chromatin structure, as well as factor binding and action, appear to work hand in hand.

## 4. Target Genes: Feedback Control by ROS and Hormones 

For WRKY53, many direct target genes turned out to be other WRKY factors pointing towards a WKRY network. Moreover, transcription factors of other families are also regulated by WRKY53, suggesting a more general role for WRKY53 in the overall senescence regulatory network [8]. Genes involved in general features of senescence as for example remobilization processes and transport are also found among the direct target genes of WRKY53. Interestingly, besides the direct feedback regulations between the WRKY factors themselves, higher order feedback loops are installed. Among the direct target genes of WRKY53 all three catalases were identified. Catalases are scavenging enzymes converting two molecules of H_2_O_2_ into water and oxygen. In this reaction, H_2_O_2_ is used as electron donor as well as electron acceptor without the need of further redox equivalents. Hence, WRKY53 influences the intracellular hydrogen peroxide level via regulation of catalase gene expression. On the other hand, H_2_O_2_ as signaling molecule can induce *WRKY53* expression [8]. The oilseed rape homolog of *WRKY53* also appears to feedback on hydrogen peroxide levels by modulating transcription of *respiratory burst oxidase homolog protein* (*Rboh*) *D* and *RbohF* [77]. Moreover, the plant hormone salicylic acid (SA) also influences the expression of many WRKY genes, including *WRKY53* as well as their DNA-binding activity [59], which clearly gives the link to pathogen response. In the case of *WRKY53*, SA has a positive whereas jasmonic acid (JA) has a negative influence on *WRKY53* expression. In general, salicylate-mediated suppression of jasmonate-responsive gene expression in *Arabidopsis* is targeted downstream of the jasmonate biosynthesis pathway. The signaling pathways of SA and JA cross-communicate providing the plant with a regulatory potential to fine-tune the outcome [78]. In *Arabidopsis*, ROS response genes appear to be activated earlier than JA response genes, which is in agreement with an early induction of *WRKY53* during monocarpic senescence in all rosette leaves and a downregulation of its expression at later time points [3,52]. Analyses of JA and SA signaling mutants revealed that JA signaling on *WRKY53* expression involves the F-box and JA receptor protein CORONATINE INSENSITIVE1 (COI1), but is independent of JASMONATE RESISTANT 1 (JAR1). On the other hand, SA signaling on the *WRKY53* promoter is influenced by the SA biosynthesis gene *SID2* but only partially mediated by the transcription factor gene *NONEXPRESSOR OF PR GENES1* (*NPR1*) [33]. A direct regulation of SA or JA biosynthesis genes by WRKY53 is not yet described; however, TCP8 and its interactor WRKY28 regulate the expression of the isochorismate synthetase gene *SID2/ISC1* [79], indicating a feedback of the WRKY network on the biosynthesis of these hormones.

## 5. Redox Regulation at the *WRKY53* Promoter

Even though the feedback regulation to control the intracellular hydrogen peroxide level through the catalase gene expression appears to be simple and clear, it is still an open question how the hydrogen peroxide signal is transmitted to the promoter of *WRKY53* leading to its induction. At least two candidates are most likely involved in this transduction, namely the WRKYs themselves, especially WRKY25, and a member of the HD-ZIPIII factor family, REVOLUTA (REV). In general, redox conditions can influence the action of transcription factors in several ways. Redox conditions can change either DNA-binding activity or activation potential. Intracellular localization or interaction with specific partners can be under the control of different redox conditions. Moreover, proteolytic degradation can be activated by changing redox conditions or even a combination of all these aspects can be in place. He at al. [80] just recently reviewed this topic very nicely. WRKY factors themselves are ideal candidates for redox regulation, as they contain one or two potentially redox-sensitive zinc-finger DNA-binding domains [81]. For one of the upstream regulators of WRKY53, namely WRKY25, a redox-dependent DNA-binding activity has been disclosed, in which oxidizing conditions dampened the action of WRKY25. Here again, a feedback control is installed. Overexpression of *WRKY25* in transgenic *Arabidopsis* plants mediated higher tolerance to oxidative stress and the intracellular H_2_O_2_ level is lower in these plants but higher in *wrky25* mutants compared to wildtype plants, suggesting that WRKY25 itself is involved in controlling intracellular redox conditions [82]. However, not all WRKYs appear to act redox-sensitive: WRKY18 does not show any functional changes under different redox-conditions [82] whereas WRKY70 binding to the W-box 2 of the *WRKY53* promoter is more efficient under oxidizing conditions (unpublished results). WRKY53 itself is also redox-sensitive but only to a low extent [82]; however, in protoplasts of *wrky53* mutant cells activation of *WRKY53* promoter-driven reporter gene expression is almost completely abolished, indicating that the WRKY53 protein itself plays a role in the H_2_O_2_ response of its own gene [72]. Furthermore, REV is also involved in the activation of *WRKY53* transcription by ROS as in the *rev* mutants the induction is dampened in a concentration-dependent manner. In this case, reducing conditions favor DNA binding [74].

## 6. DNA-Binding Activity Control beyond ROS

WRKY factors are often targeted by classical mitogen activated protein (MAP) kinase signaling, in which a MAP kinase kinase kinase (MEKKs) is activated and phosphorylates a downstream MAP kinase kinase (MEKs), which then phosphorylates a MAP kinase (MPKs) that targets transcription factors. The response to flagellin was the first identified complete plant MAP kinase cascade, in which the flagellin signal is recognized by the receptor FLS2, a leucine-rich-repeat (LRR) receptor kinase, which leads to the activation of MEKK1, MKK4/MKK5 and MPK3/MPK6, and finally of WRKY22/WRKY29 transcription factors which then activates pathogen response genes [83]. Besides the classical MAPK cascades, unusual properties of these kinases have already been described. MEKK1 can directly bind to the promoter of *WRKY53* and can increase the expression of *WRKY53*. Even though MEKK1 can directly bind to DNA, it cannot directly activate transcription, as it has no activation domain per se. Instead, MEKK1 most likely interacts with WRKY53 or other WRKYs proteins that bind to the promoter region of the *WRKY53* gene and phosphorylates them, thereby increasing or decreasing their activity [72,82]. Thus, MEKK1 is a bifunctional protein: It can bind DNA and can phosphorylate proteins. However, this also means that MEKK1 can take a short cut in the MAPK signaling and directly phosphorylates WRKY53. Furthermore, MEKK1 is also part of a feedback regulation of ROS levels. Hydrogen peroxide induces expression and activity of MEKK1, in which proteasome-mediated MEKK1 protein stabilization is most likely part of the H_2_O_2_-induced activation of MEKK1 [72,84]. On the other hand, MEKK1 activation and signaling leads to ROS accumulation in the cells [84]. This feedback loop is reminiscent of the WRKY53–H_2_O_2_ loop. As the *WRKY53* gene, *MEKK1* shows its maximum expression at the bolting and flowering time. At this time point, intracellular H_2_O_2_ levels increase leading to the activation of MEKK1, which then is involved in activation of *WRKY53* expression and in regulation of ROS levels [72,84]. Furthermore, in rice (*Oryza sativa*, *Os*) *Os*WRKY53 directly interacts with *Os*MPK3/*Os*MPK6, and *Os*WRKY53 can be phosphorylated and activated by the MPKs. At the same time, the interaction with *Os*WRKY53 leads to an inactivation of their activity indicating that *Os*WRKY53 functions as a negative feedback modulator of *Os*MPK3/*Os*MPK6 action, which allows rice plants to fine-tune their defensive investment against a chewing herbivore during early signaling [85].

Another way of influencing the activity of WRKY53 is the interaction with the EPITHIOSPECIFYING SENESCENCE REGULATOR (ESR/ESP) which is under the control of JA signaling [33]. The expression of *ESR/ESP* is antagonistically regulated to *WRKY53* in response to JA and SA, and both proteins can negatively influence the expression of the other gene. ESPs are rather small labile proteins that do not have any enzymatic activity themselves but act as cofactors for myrosinase. If EPSs are present during the hydrolysis of alkenyl glucosinolates, nitrils or epithionitriles are formed instead of isothiocyanates. Nitrils or epithionitriles confer higher resistance to *Pseudomonas syringae* and *Alternaria brassicicola* infection whereas isothiocyanates were stronger feeding inhibitors for a generalist lepidopteran herbivore, like, e.g., the cabbage looper *Trichoplusia ni*, than were nitriles [33,86]. This reaction takes place either in specialized cells (myrosin cells) or in the cytoplasm of normal cells, in which myrosinase is often associated with the membrane of the ER or the tonoplast. However, if WRKY53 protein is present, ESR/ESP can be directed to the nucleus, where ESR/ESP can inhibit DNA-binding of WRKY53, as shown by EMSA experiments [33]. This interaction between ESR/ESP and WRKYs appears to be very specific and restricted to WRKY53, as ESR/ESP can no longer enter the nucleus in *wrky53* mutant cells indicating that no other WRKY can interact with ESR/ESP and bring it into the nucleus [33].

## 7. Degradation of the WRKY53 Protein

Besides the regulation of *WRKY53* expression and activity by the diverse mechanisms mentioned above, the amount of WRKY53 protein is also tightly controlled by the HECT-domain E3 ubiquitin ligase UPL5 [87]. The UPL5 protein interacts with WRKY53 and can mark WRKY53 for 26S-proteasomal degradation by polyubiquitination. The *UPL5* gene is expressed antagonistically to the *WRKY53* gene during development as well as after treatment with JA, SA or H_2_O_2_. Accordingly, *UPL5* expression is low during bolting and flowering time when H_2_O_2_ levels increase, monocarpic senescence sets in and *WRKY53* expression is turned on in all rosette leaves. This means that the cells control WRKY53 protein levels very tightly, first by a strong regulated gene expression and, as a double bottom, by degradation of WRKY53 proteins through UPL5 and the 26S proteasome [87]. In case WRKY53 should be transiently activated during wounding or pathogen attack, UPL5 would reduce protein levels again short after induction and keep them low except during bolting and flowering time to ensure correct timing of senescence induction. 

Moreover, WRKY53 directly regulates the expression of the RING-type ubiquitin ligase, *ARABIDOPSIS TOXICOS EN LEVADURA31* (*ATL31*), which is a regulator of post-germination growth in response to changes in CO_2_/N status and is also involved in senescence regulation. Leaf senescence is accelerated in the loss-of-function mutant and suppressed in overexpressors of *ATL31* under high-CO_2_/low-N conditions [88]. Whether a feedback regulation to WRKY53 exists is still an open question; the only targets of ATL31 characterized so far are 14-3-3-proteins. CBL-Interacting Protein Kinases (CIPKs), namely CIPK14 interacting with calcineurin B-like protein 8 (CBL8), are required for ATL31 phosphorylation and stabilization leading to the degradation of 14-3-3 proteins [89]. However, some WRKYs are able to interact with 14-3-3 proteins and 14-3-3-protein are involved in senescence regulation [90] (unpublished results). Further in-depth investigations of the relationship between WRKY53 and ATL31, and of the upstream signaling cascade modulating ATL31 activity, will shed more light on the molecular mechanisms integrating the control of primary metabolism into leaf senescence. 

In addition, WRKY53 is upstream of degradation pathways acting on other WRKYs, such as WRKY57 [91]. In this case, WRKY53 directly targets the promoter of *ORE9/MAX2* [8], which encodes an F-box protein that is involved in degradation of the central brassinosteroid (BR) signal regulators BRASSINAZOLE-RESISTANT 1 (BZR1) and BRI1-EMS-SUPPRESSOR 1 (BES1) [92]. BZR1 positively regulates the expression of *WRKY57* [93], which directly represses *SAG4* and *SAG12* expression [35]. WRKY53 also binds directly to the *SAG12* promoter but induces *SAG12* expression, indicating that these two factors compete on the promoter of *SAG12* most likely for the same W-box. Moreover, ORE9/MAX2 also targets the strigolactone receptor protein D14 and, upon hormone perception, leads to the degradation of the SUPPRESSOR OF MORE AXILLARY GROWTH2-LIKE6 (SMXL6)/-7/-8-type repressor complexes [94]. Leaf senescence was strongly accelerated by the application of strigolactone but only in the presence of ethylene and not by strigolactone alone. These observations suggest that strigolactone promotes leaf senescence by enhancing the action of ethylene [95]. This would indeed again create another feedback loop, as ethylene also signals on the *WRKY53* gene. The ethylene response of the *WRKY53* gene is transmitted by ETHYLENE RESPONSE FACTOR 4 (ERF4), which regulates the expression of the WRKY53 regulator *ESR/ESP* [96]. However, an additional ESR/ESP-independent way of ERF4 signaling on the *WRKY53* promoter exists, but whether this is direct or by targeting the *CAT3* gene and thereby influencing ROS levels [97] is still an open question.

## 8. Connecting Early and Late Leaf Development

Besides all these complicated interconnections, a “leaf developmental memory” appears to exist that links early developmental processes to leaf senescence. In other words, if early leaf development is perturbed, leaf senescence is affected. The transcription factor REVOLUTA (REV) is one example illustrating this idea: REV is involved in early developmental processes, such as establishment of leaf polarity, lateral meristem initiation or vascular development, but also directly regulates the expression of *WRKY53* in a redox-dependent manner [74]. REV is mediating at least part of the redox-response of the *WRKY53* promoter as this response is dampened in *rev* mutant plants and, in addition, REV DNA-binding activity is redox-dependent [74]. In recent years, more and more examples of a cross-talk between early and late leaf development became apparent. Cell proliferation activity and leaf senescence are also interconnected, in a way that low cell proliferation activity is associated with accelerated leaf senescence and vice versa (for a review, see [98]). Obviously, when analyzing senescence, the whole life cycle of a leaf has to be taken into account. Therefore, factors that co-ordinate environmental signals as well as developmental processes throughout the plant’s life history have to be in place to regulate onset and progression of senescence to the best overall benefit of the plant.

In the last two decades, the plant circadian clock advanced from a simple timekeeper to a complex developmental manager [99] so that Sanchez and Kay already entitled the circadian clock as the “mastermind” of plant life. The evening complex (EC), which is a critical component of the core oscillator, coordinates environmental and endogenous signals [100] and represses jasmonate-induced leaf senescence [101]. In addition, CIRCADIAN CLOCK-ASSOCIATED1 regulates ROS homeostasis and oxidative stress responses through association with EC at promoters of ROS-responsive genes in vivo. Consistent with the principles of feedback regulation, ROS also affects the transcriptional output of the clock [102]. Moreover, the circadian rhythm is affected by leaf age in which the circadian periods of clock-regulated genes are shorter in older leaves compared to young leaves [103]. Therefore, the circadian clock is one of the coordinators managing plant senescence in relation to its life history.

## 9. Conclusions and Future Perspectives

The example of WRKY53 illustrates very clearly how much effort the cells are making to control expression precisely and to fine-tune activity of a single transcription factor, and we are still far from understanding this hub completely (Figure 1). Most likely, other hubs are not regulated with less complexity, and there are still many open questions. We do not know yet how the plants measure the age of individual leaves or the age of the whole plant. We do not know all signaling cascades or feedback loops completely. We still do not understand where specificity in ROS signaling comes from, as these small molecules are part of almost all signaling cascades. Are there specific signatures or is it the local production and scavenging? The combination of different “omics” approaches provided a more holistic and integrated view on the overall process and the whole plant level, and contributed to the identification of new hubs. New techniques on the single-cell level will give more inside into cell autonomous and organ-specific cues and maybe also on cell-to-cell communication. Moreover, in-depth analyses of the kinetics of the interactions will be the basis to start modeling approaches and try to simulate what will happen in different scenarios like, e.g., higher temperature, drought stress and nutrient limitations, but this will still be a long way to go.

## Figures and Tables

**Figure 1 plants-08-00578-f001:**
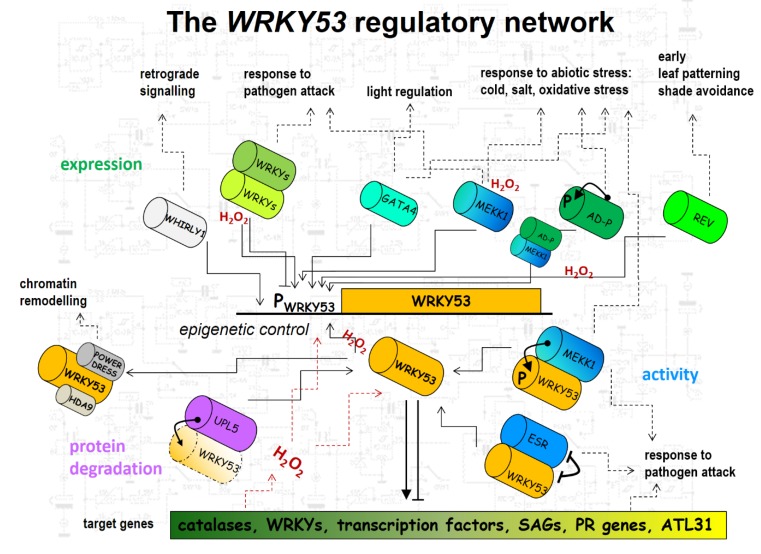
Connecting diagram illustrating the WRKY53 regulatory network. The straight black line and the orange square illustrate the *WRKY53* gene and its promoter. The orange cylinder displays the WRKY53 protein. Green cylinders describe proteins binding to the promoter and regulating the expression of *WRKY53,* including classical transcription factors like other WRKYs, GATA4 and REV or untypical DNA-binding proteins with kinase activity like AD or MEKK1. Blue cylinders indicate proteins regulating WRKY53 activity like MEKK1 and ERS/ESP. The purple cylinder represents UPL5, which is involved in degradation of WRKY53. Grey cylinders depict proteins involved in altering chromatin structure like the HDA9/POWERDRESS complex or single-strand binding proteins like WHIRLY1. Dotted lines indicate crosstalk to stress responses or other developmental processes.

**Table 1 plants-08-00578-t001:** WRKY factors differentially expressed during senescence ^1^.

WRKYs	Plant Species	References
8 *Gh*WRKYs	Cotton *Gossypium hirsutum*	[14]
54 *Gh*WRKYs	Cotton *Gossypium hirsutum*	[15]
30 *Vv*WRKYs	Grapewine *Vitis vinifera*	[16]
19 *Gj*WRKYs	Cape jasmine *Gardenia jasminoides Ellis*	[17]
9 *Hv*WRKYs	Barley *Hordeum vulgare*	[18]
23 *Pv*WRKYs	Switch grass *Panicum virgatum*	[19]
86 *Gm*WRKYs	Soybean *Glycine max*	[20]
59 *Os*WRKYs	Rice *Oryza sativa*	[21]
17 *Os*WRKYs (including *Os*WRKY40, *Os*WRKY53)	Rice *Oryza sativa*	[22]
161 *Zm*WRKYs	Maize *Zea mays*	[23]
11 *Bn*WRKYs	Oilseed rape *Brassica napus*	[24]
7 *Tp*WRKYs	Clover leaf *Trifolium pratense*	[25]
13 *Ta*WRKYs (including *Ta*WRKY7, *Ta*WRKY6, *Ta*WRKY45)	Wheat *Triticum aestivum*	[13,26]

^1^ We apologize if we missed datasets of other authors.

**Table 2 plants-08-00578-t002:** Senescence-regulating WRKYs in important model and crop plant species ^1^.

WRYK Factor	Plant Species	Overlap	References
*At*WRKY6	*Arabidopsis thaliana*	defense	[27,28]
*At*WRKY40, *At*WRKY46, *At*WRKY51, *At*WRKY60, *At*WRKY63, *At*WRKY75	*Arabidopsis thaliana*	SA signaling	[29]
*At*WRKY2, *At*WRKY18, *At*WRKY40, *At*WRKY60, *At*WRKY63	*Arabidopsis thaliana*	ABA signaling	[30]
*At*WRKY22	*Arabidopsis thaliana*	dark-induced senescence	[31]
*At*WRKY45	*Arabidopsis thaliana*	GA signaling	[32]
*At*WRKY53	*Arabidopsis thaliana*	SA, JA signaling	[8,33]
*At*WRKY54, *At*WRKY70	*Arabidopsis thaliana*	Osmotic stress, defense	[7,34]
*At*WRKY57	*Arabidopsis thaliana*	JA and auxin signaling, defense	[10,35]
*AtWRKY*75	*Arabidopsis thaliana*		[36]
*Br*WRKY6	Chinese cabbage *Brassica rapa*	GA signaling	[37]
*Br*WRKY65	Chinese cabbage *Brassica rapa*		[38]
*Ca*WRKY50	chickpea *Cicer arietinum*		[39]
11 members of the WRKY TF family (two *WRKY42* genes, *WRKY65*, *WRKY70*, *WRKY11*, three *WRKY33* genes, *WRKY41*, *WRKY30*, and an unknown WRKY domain-containing protein)	Lemon *Citrus sinensis* (L.)	Postharvest, fruit senescence	[40]
*Gh*WRKY17	Cotton *Gossypium hirsutum*		[41]
*Gh*WRKY27	Cotton *Gossypium hirsutum*		[42]
*Gh*WRKY42	Cotton *Gossypium hirsutum*		[43]
*Gm*WRKY53b	Soybean *Glycine max*	Blue light signaling	[44]
*Hv*WRKY12	Barley *Hordeum vulgare*		[45]
*Os*WKRY42	Rice *Oryza sativa*	ROS	[46]
*Os*WRKY13	Rice *Oryza sativa*		[47]
*Os*WRKY14	Rice *Oryza sativa*		[48]
*Os*WRKY80	Rice *Oryza sativa*	Fe-response, drought	[49]
*Zm*WRKY20, *Zm*WRKY36, *Zm*WRKY50, *Zm*WRKY71	Maize *Zea mays*		[50]

^1^ We apologize if we missed datasets of other authors.

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
