# Peer review of "Arabidopsis WRKY53, a Node of Multi-Layer Regulation in the Network of Senescence"

_plants, 2019, doi:10.3390/plants8120578_

Round 1

Reviewer 1 Report

In this manuscript by Zentgraf and Doll, the authors summarize the molecular mechanisms of leaf senescence focusing on WRKY transcription factors. This paper is well-written and contributes to our understanding of leaf senescence.

I listed up minor points below.

Line 190: “2” in “H2O2” should be written as subscript.

Line 190: “wrky25” should be written in italics.

Line 250: “UPL5” should be written in italics?

Table 1: For 59 OsWRKYs, “10.1038/srep20881” should be deleted?

Author Response

Thank you very much for your positive response.

All points have been corrected accoding to the advice

Line 190: “2” in “H2O2” should be written as subscript.

Done

Line 190: “wrky25” should be written in italics.

Done

Line 250: “UPL5” should be written in italics?

Done

Table 1: For 59 OsWRKYs, “10.1038/srep20881” should be deleted?

Deleted.

Reviewer 2 Report

WRKY transcription factors comprise one of the largest families in plants. These proteins contain highly conserved WRKY domain that binds to the equally conserved binding site, so-called the W box (TTGACC/T). Initially, their role as positive or negative regulators of the plant immune response was indicated. Later on, WRKY TFs were shown to play an essential role in the regulation of many plant processes, such as responses to environmental clues and the expression of developmental programs. And, what is extremely important, WRKY TFs was shown to regulate the expression of other TFs including these from their own family as well as bZIPs, MYBs and ERFs. It suggests that the initial division of genes into TFs and effector genes is an over-generalization. The first level of complications comes from the interaction of different TFs on the promoters of individual genes. But the real level of intrinsic complexity becomes to light only now and WRKY53 functioning during senescence is a good example. Reviewed manuscript greatly illustrates the intricacy of WRKY53 inducible expression, post-transcriptional regulation of protein stability and network of activated target genes. The manuscript also includes the reference to the regulation of WRKY53 network activity by the physiological state of the cell. It is a valuable summary of the current state of knowledge and is worth publishing.

line 544 – add full reference (now there is "82. Doll et al.")

Author Response

thank you very much for your positive feedback.

line 544 – add full reference (now there is "82. Doll et al.")

The full reference has been added. Unfortunately, the final decision is still pending but the manuscript has already been provitionally accepted by the reviewers and Guest Editor. The final decision of the Editorial Office is still missing. This will be updated as soon as possible.

Reviewer 3 Report

The review on the WRKY53 regulatory network in senescence by Ulrike Zentgraf and Jasmin Doll, that is very well written and interesting to a wide audience.

It is almost ready to be published, there are only few typos to correct like "extends" instead of "extent", or other little things that the editorial office can manage before publication, few sentences that are repeated (line 104-105) or unclear (line 282-284), but I think it can be accepted right away after a final polish before it gets published.

Author Response

Thank you very much for your positive judgement on our review.

Several typos have been removed.

Repetition (line 104-105): this has been rephrased.

Unclarity (line 282-284): This has also been rephrased to provide more clarity.